# Competitive Analysis of Rumen and Hindgut Microbiota Composition and Fermentation Function in Diarrheic and Non-Diarrheic Postpartum Dairy Cows

**DOI:** 10.3390/microorganisms12010023

**Published:** 2023-12-22

**Authors:** Yangyi Hao, Tong Ouyang, Wei Wang, Yajing Wang, Zhijun Cao, Hongjian Yang, Le Luo Guan, Shengli Li

**Affiliations:** 1State Key Laboratory of Animal Nutrition, Beijing Engineering Technology Research Center of Raw Milk Quality and Safety Control, College of Animal Science and Technology, China Agricultural University, Beijing 100193, China; haoyangyi0928@163.com (Y.H.); caozhijun@cau.edu.cn (Z.C.); yang_hongjian@cau.edu.cn (H.Y.); 2Department of Agricultural, Food and Nutritional Science, University of Alberta, Edmonton, AB T6G 2P5, Canada

**Keywords:** postpartum cows, nutritional diarrhea, gastrointestinal fermentation, 16S rRNA

## Abstract

Postpartum dairy cows can develop nutritional diarrhea when their diet is abruptly changed for milk production. However, it is unclear whether nutritional diarrhea develops as a result of gut acidosis and/or dysbiosis. This study aimed to uncover changes in the gastrointestinal microbiota and its fermentation parameters in response to nutritional diarrhea in postpartum dairy cows. Rumen and fecal samples were collected from twenty-four postpartum cows fed with the same diet but with different fecal scores: the low-fecal-score (LFS: diarrheic) group and high-fecal-score (HFS: non-diarrheic) group. A microbiota difference was only observed for fecal microbiota, with the relative abundance of *Defluviitaleaceae_UCG-011* and *Lachnospiraceae_UCG-001* tending (*p* < 0.10) to be higher in HFS cows compared to LFS cows, and *Frisingicoccus* were only detected in HFS cows. The fecal bacterial community in LFS cows had higher robustness (*p* < 0.05) compared to that in HFS cows, and also had lower negative cohesion (less competitive behaviors) and higher positive cohesion (more cooperative behaviors) (*p* < 0.05) compared that in to HFS cows. Lower total volatile fatty acids and higher ammonia nitrogen (*p* < 0.05) were observed in LFS cows’ feces compared to HFS cows. The observed shift in fecal bacterial composition, community networks, and metabolites suggests that hindgut dysbiosis could be related to nutritional diarrhea in postpartum cows.

## 1. Introduction

After calving, the introduction of a high-grain diet can cause a maladaptive issue of the gastrointestinal tract in postpartum dairy cows [1], and as a result, cows can develop nutritional diarrhea [2]. Although many cows can recover eventually, some of them need to be treated or culled due to the severity of nutritional diarrhea. Diarrhea in dairy cows can be caused by both pathogenic infections and host-related factors [2,3]. The major pathogenic factors include the virulence of the pathogens, the types of infectious microbes, such as bacterial, viral, and protozoal pathogens, and the number of concurrent infectious diseases [3]. The host factors include nutrition, immunity to specific pathogens, general health, and age upon exposure to pathogens [2,3]. Among host-related non-pathogen-caused diarrhea, nutritional diarrhea is well studied in monogastric animals. For example, an imbalance between diet composition and gut microbial community is a major cause of nutritional diarrhea in weaning piglets [4]. Recent evidence has indicated that imbalanced diets can alter the commensal bacterial composition in the lower gut and reduce short-chain fatty acid (SCFA) production, which can inhibit pathogens and supply energy sources for gut epithelium cells [4,5]. Additionally, the excess fermentation of protein by hindgut bacteria can produce various toxic metabolites, such as ammonia nitrogen (NH_3_-N) and amines, which can promote the growth of potential pathogens in the gut [5], leading to a higher incidence of nutritional diarrhea [4]. However, the mode of action of nutritional diarrhea in postpartum cows is not clearly defined.

Hindgut bacteria and their fermentation byproducts are essential in colonic inflammatory response regulation [6]. Compared to the rumen, the hindgut epithelium is ostensibly more susceptible to lesions in response to the dysbiosis of microbial fermentation because it lacks the reinforcement of four multilayered strata of the epithelium [1]. It has been reported that dysbiosis of the hindgut microbial community can down-regulate intestinal barrier gene expression and promote inflammatory factor gene expression in weaning goats [6]. Although rumen bacteria can not cause diarrhea directly, they can affect the nutrient substrate flow to the lower gut and shape the lower gut environment indirectly [1]. For example, a high-grain diet can lead to damage to the mucus layer in lactating goats [7] and subacute ruminal acidosis (SARA) [8] in postpartum dairy cows. As a result, cows fed a high-grain diet have increased lipopolysaccharides (LPS) in their gastrointestinal tract and blood [9] that can further stimulate the production of nitric oxide and prostaglandin E_2_, and then, lead to diarrheagenic activity [10]. However, it is unclear whether nutritional diarrhea in postpartum dairy cows is related to dysbiosis of the rumen and hindgut microbiota and the shift in LPS and/or SCFA productions.

The hypothesis is that the sudden introduction of a high-energy, high-protein diet to postpartum dairy cows could lead to dysbiosis in the microbiota of the rumen and hindgut, impacting their metabolites and resulting in nutritional diarrhea. Specifically, the abrupt increase in a high-grain diet might elevate lipopolysaccharide (LPS) levels and alter microbial protein degradation and fermentation in the gut, which further lead to nutritional diarrhea in postpartum dairy cows. In this study, cows exhibiting high or low fecal scores (indicative of nutritional diarrhea) were selected, and their milking performance; their ruminal and fecal fermentation profiles; their ruminal, fecal, and blood LPS levels; the apparent total tract digestibility of their feed; as well as the composition of the rumen and fecal bacteria were compared. The aim was to elucidate the factors contributing to nutritional diarrhea in postpartum dairy cows on commercial dairy farms.

## 2. Materials and Methods

### 2.1. Animal Management and Sample Collection

The experiment was conducted at a commercial dairy farm named Hefei Dairy Farm in Anhui province, China (31°93′ S, 117°68′ W), from November to December 2020. All the procedures carried out during this experimental period were approved by the China Agriculture University Laboratory’s Animal Welfare and Animal Experimental Ethical Faculty (protocol number: AW81102202-1-2). All cows were fed three times (07:30, 14:30, and 20:30) and had free access to water throughout the day. A total mixed ration (TMR) was supplied to cows, which was a whole-plant corn silage and soybean meal-based diet with a crude protein (CP) percentage of 18.08%, neutral detergent fiber (NDF) of 31.65%, and starch of 25.61% (dry matter base) (Appendix A), and met the requirements of a cow with a body weight of 700 kg producing 35 kg/d of milk [11]. In all experimental cows, we ruled out bovine viral diarrhea, paratuberculosis, and parasites following the commercial farm standard operating procedure. The fecal score was evaluated following the standard in Cow Signals [12] as follows: 1: cream soup consistency; 2: does not stack and a less than 1 inch thickness; 3: consistency similar to thick pancake batter and exhibiting a slight divot in the middle; 4: thick and starting to become somewhat deeper, yet does not stack; and 5: extremely well formed, firm, and stacks over 2 inches in thickness. Twenty-four multiparous Holstein dairy cows were enrolled in this study, with similar body weights (637 ± 28 kg, mean ± standard error), ages (5.34 ± 0.27, years), body condition scores (2.78 ± 0.19), and differences in fecal scores. The experimental period started after calving and ended at 19 days in milk. And then, the cows were divided into two groups, low fecal score (LFS, *n* = 12) and high fecal score (HFS, *n* = 12), in the middle of the experimental period (d9). Fecal water content was also measured with a forced-air oven (DGG-9240B; Shanghai-ShenXin Inc., Shanghai, China) at 60 °C for 48 h to further validate the fecal score accuracy.

### 2.2. Sample Collection

Milk samples were collected on d9 after calving at 07:00, 14:00, and 20:00, and pooled with a 4:3:3 ratio for the milk composition test. The rumen fluid was collected an hour before the morning feed using an oral gastric tube (Ancitech, Winnipeg, MB, Canada) on d9 after calving. The initial 50 mL rumen fluid collected was discarded to avoid saliva contamination, and the next 50 mL rumen fluid collected was processed for downstream analysis. Briefly, about 5 mL rumen fluid was centrifuged at 13,000× *g* for 40 min at 4 °C; then, the supplement was collected and filtered into a pyrogen-free glass tube and heated at 100 °C for 30 min; finally, the cooled supplement was stored at −20 °C for subsequent LPS detection [13]. The remaining unprocessed rumen fluid was stored at −80 °C for DNA extraction and fermentation profile analysis. Fecal samples were collected from the rectum at 07:30, 14:30, and 20:30 on three consecutive days from d9 to d12 after calving. About 300 g of feces was collected each time and stored at −20 °C for apparent total tract digestibility (ATTD) detection. For the fecal sample collected at 07:30 on d9, an extra 20 g fecal sample was collected and equally separated into two parts. One part was mixed with the same amount of physiological saline (0.90% *w*/*v* of NaCl) and immediately centrifuged at 3000× *g* for 15 min, and then, the supernatants were stored at −20 °C until they were analyzed for volatile fatty acid (VFA) and NH_3_-N [9]. The other 10 g fecal sample was transferred into a pyrogen-free tube with 10 mL of physiological saline and fully mixed for LPS measurement [14]. Blood samples were collected from the tail vein using a no-anticoagulation blood collection tube (Vacutainer; Becton Dickinson, Nanjing, China) before morning feeding on d9 after calving. Blood samples were centrifuged at 3500× *g* at 4 °C for 15 min to obtain serum, and then, stored at −20 °C for further analysis. TMR was also collected at each feeding time on d9 and finally composited to obtain one sample, which worked as the intrinsic digestibility marker for the measurement of feed ATTD.

### 2.3. Milk Yield, Composition, and Feed Digestibility Measurements

The milk yield was recorded using automatic milk flow equipment (E300, DeLaval International AB, Tumba, Sweden). Milk fat, protein, lactose, total solids (TS), milk urea nitrogen (MUN), somatic cell count (SCC), BHBA, and acetone on d9 were measured using an automated near-infrared milk analyzer (CombiFoss FT+; Foss Electric, Hillerød, Denmark) at the Nanjing Dairy Herd Improvement Testing Center (Nanjing, China).

About 25% of the well-mixed fecal sample in each cow and TMR sample were dried and milled using a feedstuff mill (KRT-34; KunJie, Beijing, China) with a 1 mm screen for downstream chemical composition analysis. NDF and acid detergent fiber (ADF) of fecal and TMR samples were analyzed using an ANKOM fiber analyzer added with heat-stable α-amylase and sodium sulfite (A2000i; American ANKOM, Macedon, NY, USA), referencing Van Soest et al. [15]. The ether extract (EE) was measured by method 973.18 of the Association of Official Analytical Chemists [16]. CP was calculated by multiplying 6.25 by the concentration of nitrogen, which was detected with method 968.06 [16]. Organic matter (OM) was determined by calculating 100% minus the ash content, and the ash was detected in a muffle furnace (TL0610; KeAo, Beijing, China) at 550 °C for 6 h with method 924.05 [16]. Acid-insoluble ash (AIA) as a digestibility marker was determined according to the method described by [17]. The AIA concentration in the feces (*Af*, g/kg) and diet (*Ad*, g/kg) and the concentrations of OM, NDF, ADF, EE, and CP in the feces (*Nf*, g/kg) and diet (*Nd*, g/kg) were used to calculate the ATTD, which was referenced from Schneider and Flatt [18], using the following formula:(1)$$ATTD(\%)=\left[1-\frac{Ad\times Nf}{Af\times Nd}\right]\times 100$$

### 2.4. Ruminal and Fecal Fermentation Profiles and LPS Measurements

Rumen fluid pH was measured immediately after sample collection with a pH electrode (model pH B-4; Shanghai Chemical, Shanghai, China). The ruminal fluid and fecal NH_3_-N concentration were measured using the phenol–sodium hypochlorite colorimetry method on a spectrophotometer (721, INESA analytical instrument Co., Ltd., Shanghai, China), as described by Broderick and Kang [19]. The VFAs of rumen fluid and fecal were measured using gas chromatography (6890 N; Agilent technologies, Avondale, PA, USA) with a capillary column (0.32 mm × 0.50 mm film thickness) following the methods described by Cao et al. [20].

The ruminal fluid and fecal supernatant LPS were diluted at 1:100,000 and measured using a chromogenic end-point Tachypleus amebocyte lysate assay kit (Chinese Horseshoe Crab Reagent Manufactory, Xiamen, China) with a minimum detection limit of 0.01 to 1 endotoxin units EU/mL, with reference to Liu et al. [21]. The serum LPS was detected using the kit described above. Serum lipopolysaccharide-binding protein (LBP) was measured according to the manufacturer’s guidelines for the commercial kit (Nanjing Jiancheng Bioengineering Institute, Nanjing, China).

### 2.5. Rumen and Hindgut Microbiota Composition and Network Analysis

#### 2.5.1. DNA Extraction and Amplicon Sequencing

Total DNA was extracted from rumen fluid and fecal samples using the PowerSoil DNA Isolation Kit (MoBio Laboratories, Carlsbad, CA, USA) following the manufacturer’s guidelines. The DNA quantity, purity, and integrity were determined using a NanoDrop 2000 spectrophotometer (Thermo Fisher Scientific, Waltham, MA, USA) and 1% agarose gel electrophoresis. The primer pairs 338f (5′-ACTCCTACGGGAGGCAGCAG-3′) and 806R (5′-GGACTACHVGGGTWTCTAAT-3′) targeting the bacteria partial 16S rRNA gene were used to produce amplicons for each sample; the amplicons, with an equal amount from each sample, were pooled; and the Illumina PE Miseq 300 platform was used for sequencing to obtain 300 bp paired-end reads.

#### 2.5.2. Sequence Data Analysis

The raw sequence data were processed using QIIME2 (Version 2022.2) [22]. Quality control, denoising, the removal of chimeric sequences, and the generation of amplicon sequencing variants (ASVs) were performed using the QIIME2 plugin DADA2 [23]. Taxonomy was evaluated with the feature classifier command in QIIME2 using ASVs against the SILVA database (Version 132). The adequacy of sequencing depth was evaluated by Good’s coverage index. Alpha diversity (Shannon: richness and Chao1: evenness) and beta diversity (Weighted UniFrac distance matrix) were calculated using the scripts implemented in QIIME2 with a depth of 8509, which was the lowest ASV number in the dataset. Microbial taxa detected in at lowest third of the samples were included in the downstream analysis.

#### 2.5.3. Microbiota Network Construction

The ruminal and fecal bacterial networks were constructed on the basis of Pearson correlations of log-transformed ASV abundances, followed by a random matrix theory (RMT)-based approach that determines the correlation cut-off threshold automatically [24]. This RMT-based network tool is named the Molecular Ecological Network Analysis Pipeline (MENAP) and is available at the Institute for Environmental Genomics, University of Oklahoma (http://ieg4.rccc.ou.edu/MENA/ accessed on 28 September 2022). Various network topological indices (edge number, node number, average degree, and average distance) were calculated in the MENAP interface to characterize the topological structure of the bacterial community networks.

Robustness was calculated to test the network stability on the basis of simulation [25,26]. For simulations of random species removal, a certain proportion of nodes was randomly removed. For the targeted removal simulations, certain numbers of module hubs were removed. The proportion of the remaining nodes was reported as the network robustness. We measured the robustness when 50% of the random nodes or three module hubs were removed. To determine the degree of competitive interactions or cooperative behaviors of ruminal and fecal bacterial communities in cows with different fecal scores, the negative and positive cohesions in each sample were calculated based on the abundance-weighted, null model-corrected positive and negative correlations, respectively [27]. Negative cohesion, derived from negative pairwise correlations, could reflect the degree of competitive behaviors among the community members. Positive cohesion, derived from positive pairwise correlations, could reflect the degree of cooperative behaviors among the community members.

### 2.6. Statistics

The data of milk yield, composition, gastrointestinal fermentation profiles, ATTD, LPS, network robustness, and cohesion were analyzed by Student’s *t*-test based on their normality and homogeneity of variance in SAS (version 9.4, SAS Institute Inc., Cary, NC, USA). Bacteria α diversity indices and the relative abundance of ruminal and fecal bacterial genera were analyzed via the Wilcoxon rank-sum test and the *p*-value was adjusted using false discovery rate correction, as described by [28]. The fecal bacteria β diversity was analyzed by permutational multivariate ANOVA (PERMANOVA) using the R package vegan (v2.5.6) in R studio (4.1.0). The correlation between bacterial genera and fecal water content, rumen and fecal fermentation profiles, and APPT was analyzed by Spearman rank correlation in SAS. A *p*-value ≤ 0.05 was considered significant, and a tendency of 0.05 < *p*-value ≤ 0.10 was considered for all statistic analyses.

## 3. Results

### 3.1. Rumen and Hindgut Fermentation Profiles and LPS Concentrations

As we observed, the fecal score in diarrheic cows was lower than in non-diarrheic cows (*p* < 0.001). Meanwhile, the fecal water content in LFS cows was significantly higher than that in HFS cows (*p* < 0.001) (Figure 1). The rumen pH, NH_3_-N, and TVFA concentrations did not differ between the two groups (Table 1). The ruminal molar proportion of propionate in LFS cows was higher than that in HFS cows (*p* < 0.05), while the molar proportion of acetate and A/P ratio in LFS cows was lower than those in HFS cows (*p* < 0.05). The fecal TVFA concentration and A/P ratio in LFS cows were lower than those in HFS cows (*p* < 0.05). However, the fecal NH_3_-N concentration, isobutyrate, and isovalerate molar proportions were higher in the LFS group than those in the HFS group (*p* < 0.05). No difference in fecal and ruminal fluid, serum LPS, and serum LBP concentrations were observed between the two groups (Table 2). The ATTD of NDF, ADF, EE, and OM showed no difference between LFS and HFS cows (Appendix A), while the ATTD of CP in the LFS group tended to be lower than that in the HFS group (*p* < 0.10).

### 3.2. Ruminal and Fecal Bacterial Composition and Diversity

A total of 3,079,456 raw reads were obtained with an average of 65,270 ± 8077 (mean ± standard deviation) reads per sample. After quality control, 756,358 high-quality reads remained with an average of 15,623 ± 2217 reads per sample. Good’s coverage was >99% for each sample, indicating that the sequence depth was sufficient to represent the bacterial community composition. After taxonomy classification, 15 phyla (99.9% of reads), 69 families (94.9% of reads), and 162 genera (65.5% of reads) were identified. No significance was observed in the ruminal and fecal bacteria Chao1 and Shannon indices (Figure 2). The fecal β diversity, which was calculated with weighted UniFrac distance, was significantly different (*p* = 0.048) between HFS and LFS cows. However, no significant difference was observed in ruminal bacteria β diversity (both weighted and unweighted UniFrac distance) and unweighted UniFrac distance-based β diversity in the fecal bacterial community. The distribution of all the samples did not separate obviously in the principal coordinate analysis between HFS and LFS cows in ruminal and fecal bacterial communities. 

No significant difference was observed in the relative abundance of ruminal bacterial genera between the two groups (Appendix A). The relative abundance of *Cellulosilyticum*, *Defluviitaleaceae_UCG-011*, and *Lachnospiraceae_UCG-001* tended to be higher in HFS cows’ feces (*P*adj < 0.10), and *Frisingicoccus* and *Intestinibacter* were only detected in HFS cows’ feces. The relative abundance of *Lachnospiraceae_NK3A20_group* tended to be lower in HFS cows’ feces than in that of LFS cows (*P*adj = 0.09).

### 3.3. Relationship between Rumen and Fecal Bacteria and Fecal Water Content and Fermentation Profiles

We further explored the relationship between rumen and fecal bacteria with cows’ phenotypes, and a total of four ruminal bacterial genera had a significant correlation with fecal water content (Figure 3A). Among these bacterial genera, *Clostridia_UCG-014* (r = −0.452, *p* = 0.012) and *Lachnospiraceae_NK3A20_group* (r = −0.451, *p* = 0.016) were negatively correlated with fecal water content, while *Oribacterium* (r = 0.435, *p* = 0.010) and *Prevotellaceae_UCG-004* (r = 0.423, *p* = 0.022) were positively correlated with fecal water content. There were seven fecal bacterial genera significantly correlated with fecal water content (Figure 3B). Among them, *Bifidobacterium* (r = −0.409, *p* = 0.025), *Cellulosilyticum* (r = −0.707, *p* = 0.001), *Defluviitaleaceae_UCG-011* (r = −0.528, *p* = 0.004), *Frisingicoccus* (r = −0.480, *p* = 0.015), and *Lachnospiraceae_UCG-001* (r = −0.406, *p* = 0.019) were negatively correlated with fecal water content, while *Lachnospiraceae_NK3A20_group* (r = 0.430, *p* = 0.021) and *Mogibactterium* (r = 0.469, *p* = 0.025) were positively correlated with fecal water content.

The bacterial genera that had a significant correlation with fecal water content were further analyzed to determine their relationships with gastrointestinal fermentation parameters and apparent digestibility. The ruminal bacterial genus *Oribacterium* was negatively correlated with rumen acetate (r = −0.489, *p* = 0.032) and A/P ratio (r = −0.517, *p* = 0.019), while it was positively correlated with ruminal propionate (r = 0.504, *p* = 0.013) (Figure 4). The ruminal bacterial genus *Prevotellaceae_UCG-004* was positively correlated with rumen propionate (r = 0.477, *p* = 0.011) and negatively correlated with rumen A/P ratio (r = −0.454, *p* = 0.021). The fecal bacterial genus *Bifidobacterium* was negatively correlated with fecal propionate (r = −0.657, *p* < 0.001), NH3-N (r = −0.478, *p* = 0.015), isobutyrate (r = −0.481, *p* = 0.026), and isovalerate (r = −0.415, *p* = 0.033), while it was positively correlated with A/P ratio (r = 0.644, *p* < 0.001). The fecal bacterial genus *Defluviitaleaceae_UCG-011* was positively correlated with DCP (r = 0.599, *p* = 0.003) and DOM (r = 0.504, *p* = 0.011). The fecal bacterial genus *Frisingicoccus* was negatively correlated with propionate (r = −0.631, *p* < 0.001), NH_3_-N (r = −0.641, *p* < 0.001), isobutyrate (r = −0.581, *p* = 0.002), and isovalerate (r = −0.481, *p* = 0.022) and positively correlated with A/P ratio (r = 0.586, *p* = 0.004). The fecal bacterial genus *Lachnospiraceae_UCG-001* was negatively correlated with propionate (r = −0.459, *p* = 0.013), NH_3_-N (r = −0.448, *p* = 0.024), isobutyrate (r = −0.593, *p* = 0.003), and isovalerate (r = −0.536, *p* = 0.002). The fecal bacterial genus *Mogibacterium* was negatively correlated with fecal acetate (r = −0.444, *p* = 0.027) and A/P ratio (r = −0.517, *p* = 0.019), while it was positively correlated with propionate (r = 0.451, *p* = 0.021), isobutyrate (r = 0.418, *p* = 0.026), and isovalerate (r = 0.466, *p* = 0.011).

### 3.4. Difference in Rumen and Fecal Bacterial Community Networks and Topologies between Two Groups of Cows

The ruminal and fecal bacterial community networks in HFS cows were visually less complicated than those in LFS cows (Figure 5A). The ruminal and fecal bacterial community had a higher edge number (rumen: 563 vs. 303 and fecal: 276 vs. 230), node number (rumen: 302 vs. 194 and fecal: 215 vs. 171), and average distance (rumen: 4.84 vs. 3.211 and fecal: 5.96 vs. 5.38) in LFS cows compared to that in HFS cows (Figure 5B). The ruminal bacterial community network average degree in LFS cows was higher than that in HFS cows (3.73 vs. 3.12), while the fecal bacterial community network average degree in LFS cows was lower than that in HFS cows (2.65 vs. 2.69). The ruminal bacterial community had higher robustness (*p* < 0.001) in HFS cows compared with that in LFS cows under random and targeted removal conditions, while the fecal bacterial community had lower (*p* < 0.05) robustness in HFS cows than in LFS cows (Figure 6). No significant difference was observed in ruminal bacteria negative cohesion between LFS and HFS cows, while it was significantly higher (*p* < 0.012) in HFS cows’ fecal bacterial community compared to that in LFS cows. The positive cohesion in HFS cows’ ruminal bacterial community was higher (*p* = 0.033) than that in LFS cows, while it was lower (*p* = 0.026) in HFS cows’ fecal community compared with that in LFS cows.

## 4. Discussion

In ruminants, the over-fermentation of carbohydrates under a high-energy diet in the gastrointestinal tract can lead to acidosis, and as a result, it can cause diarrhea [2]. No difference in ruminal pH suggests that ruminal acidosis was not detectable in all cows enrolled in this study. Additionally, LPS is the main detrimental product when ruminants suffer gastrointestinal acidosis [9]. However, no difference in ruminal, fecal, and serum LPS between HFS and LFS cows suggests that diarrhea in the cows was not caused by LPS as a result of gastrointestinal acidosis [29,30]. In the current study, we found that cows with nutritional diarrhea had a significant shift in their fecal fermentation profiles, bacterial composition, and bacterial networks, suggesting that potential hindgut microbial dysbiosis could be the main factor contributing to this health problem in postpartum dairy cows.

The lower TVFAs and higher NH_3_-N in the feces of LFS cows were consistent with the findings reported previously, which state that the hindgut microbes and their potential functions can be shifted more toward fatty acids synthesis and less toward deamination under high-grain diet conditions in healthy dairy cows [31]. Therefore, less carbohydrate fermentation (lower TVFAs) and more deamination activity (higher NH_3_-N) in LFS cows could be the result of dysbiosis of the hindgut microbial community and fermentation under the high-grain diet. The bacteria *Cellulosilyticum*, *Defluviitaleaceae_UCG-011*, and *Lachnospiraceae_UCG-001* tended to be higher in the feces of HFS cows. These bacterial genera have been identified as carbohydrates utilizer and positively correlated with carbohydrate metabolism [31,32], which may contribute to the observed higher TVFA concentration in HFS cows compared to LFS cows. The lower fecal TVFA in LFS cows was also similar to the findings in monogastric animals, which state that fecal TVFA was lower in the diarrheic group than in the healthy one [33,34,35]. Additionally, we found the fecal acetate concentration in HFS cows was higher than that in LFS cows. Acetate has been reported to protect the intestine from enteropathogenic infection by inhibiting the translocation of toxins from the gut lumen to the blood [36]. Although we did not detect whether the LFS cows had any pathogen infections, the observed dysbiosis and lower acetate concentration suggest that the dysbiosis in LFS cows could lead to them being more susceptible to pathogen infections.

Although no significant difference was observed in the relative abundance of fecal *Bifidobacterium* between LFS and HFS cows, the significant correlations between its relative abundance with fecal water content, NH_3_-N, and acetate concentration suggest *Bifidobacterium* could play an important role in maintaining hindgut fermentation homeostasis in our study. As a well-known probiotic organism, *Bifidobacterium* can produce acetate [37] and reduce gut NH_3_-N concentration [38], suggesting that it can contribute to a balanced environment in a cow’s hindgut, and it is important to maintain its population in the gut postpartum. The *Frisingicoccus* genus belongs to the *Lachnospiraceae* family, which has been reported as an SCFA-producing member and is positively related to gut health [39]. No *Frisingicoccus* was detected in the LFS cows, suggesting that a lack of this bacterial genus could lead to detrimental effects on the gut health of dairy cows. In the meantime, a negative correlation was found between *Frisingicoccus* and fecal NH_3_-N concentration. A high NH_3_-N concentration can damage the hindgut epithelium [40], which was found to be harmful to water absorption in the hindgut [4]. Meanwhile, when animals suffer from diarrhea, their gut epithelial structure can be damaged, which can directly affect the digestive enzymes’ secretion and nutrient absorption [4], especially for the digestion and absorption of protein in the small intestine. This can cause diarrhea and, at the same time, contribute to lower CP ATTD in LFS cows compared to HFS cows. All of these suggest that the presence of *Frisingicoccus* in the gut of HFS cows could play an important role in affecting gut epithelial function by affecting or lowering the NH_3_-N concentration. However, the function of members of this genus in the gut of dairy cows has not been well studied, so further study is required to determine the functions of this genus in ruminants’ hindguts.

Although our analysis revealed that four ruminal bacterial genera were significantly correlated with fecal water content, it is unknown whether these bacteria contributed to diarrhea or not. Among them, *Oribactrium* and *Prevotellaceae_UCG-004* were positively correlated with ruminal propionate in our study. Ruminal propionate was highly positively correlated with starch fermentation and feed fermentation efficiency [41], and the higher propionate in HFS cows suggests more efficient starch fermentation in the rumen. A recent study suggests that ruminal function can reflect the lower gut’s health status by producing extra organic acids [42]. Further studies are required to measure whether there is a difference in the microbial metabolites that transfer from the rumen to hindgut, which could trigger dysbiosis in the lower gut of LFS dairy cows.

In addition to the gastrointestinal bacterial composition, we also found that microbial interactions differed between the two groups of cows, which may also contribute to nutritional diarrhea. Microbial networks were more complex in LFS cows compared to HFS cows in regard to their network topology. An altered gut microbial network has been reported to be associated with diarrhea in calves [43]. The significant difference in cohesion values indicated that there was a distinct difference in the microbial competitive and/or cooperative behaviors between the two groups of cows in our study. A more competitive community (higher negative cohesion) can be more susceptible to trophic shift, while a less cooperative community (lower positive cohesion) can be less resilient to trophic change [44]. Therefore, the higher negative cohesion and lower positive cohesion in HFS cows suggests that their hindgut microbial community is more adaptable to the abrupt dietary shift after calving. Furthermore, we observed there was higher robustness in LFS cows’ fecal bacterial community compared to HFS cows. This higher robustness value indicated that the microbial community network was more stable [45], and it could have negative effects on remodeling the bacterial community to adapt to new trophic structures [31,46]. Therefore, the higher robustness value in LFS cows’ hindgut bacterial community indicated that the transition for their hindgut microbiota to new diet can be more challenging. In contrast to the fecal bacterial community, the ruminal bacterial community in HFS had a higher cooperative degree (higher positive cohesion) and robustness, suggesting that the rumen microbial community of HFS cows is more stable in the dietary shift and maintains stable rumen fermentation. This can lead to lower nutrient flow to the hindgut and reduced pressure for the hindgut microbiota under the pulse of dietary shift.

## 5. Conclusions

This study revealed differences in the gastrointestinal fermentation profiles, as well as the gastrointestinal bacterial composition, between diarrhetic and non-diarrhea postpartum dairy cows. The lack of a difference in ruminal pH, total volatile fatty acids, and lipopolysaccharide indicates that nutritional diarrhea in the postpartum dairy cows was not caused by rumen acidosis and/or lipopolysaccharides. The lower total fecal volatile fatty acids and higher ammonia nitrogen concentration in low-fecal-score cows suggest that nutritional diarrhea in the postpartum dairy cows in this study could have been caused by the low carbohydrate and high protein fermentation in the lower gut as well as gut dysbiosis. Key fecal bacterial genera such as *Bifidobacterium*, *Frisingicoccus*, and *Lachnospiraceae_UCG-001* were identified as significantly correlated with fecal water content and fermentation profiles, which could maintain homeostasis in the hindgut. Besides, the low-fecal-score cows’ fecal bacterial community had a lower adaptability to the diet shift compared to high-fecal-score cows, which could affect the microbial composition and fermentation in the hindgut. Metagenomes and metabolomes are required in the future to assess the microbial function and metabolites in the gut to uncover the mechanism of nutritional diarrhea in postpartum dairy cows. Regardless, the findings from this study provide knowledge on the importance of shaping hindgut bacterial composition and function in the lower gut of postpartum dairy cows, which could be one of the solutions to improving gut health and reducing nutritional diarrhea in postpartum dairy cows.

## Figures and Tables

**Figure 1 microorganisms-12-00023-f001:**
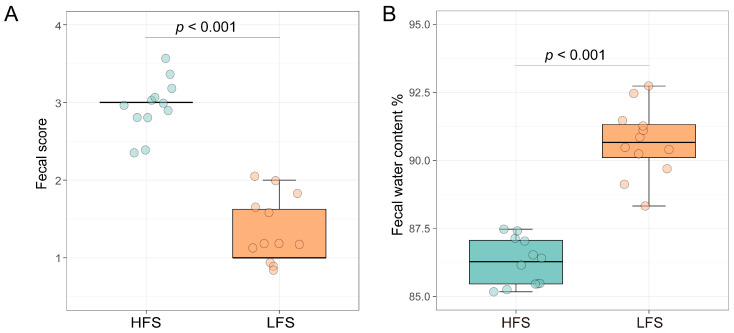
Fecal score (**A**) and water content (**B**) of dairy cows with diarrheic and non-diarrheic phenotypes. LFS: low fecal score (*n* = 12); HFS: high fecal score (*n* = 12). Boxes represent the interquartile range (IQR) between the first and third quartiles, and the horizontal line inside the box represents the median. Whiskers represent the lowest and highest values within 1.5 times the IQR from the first and third quartiles, respectively. The green color represents the HFS group and the orange colour represents the LFS group, and the circles indicate each cow’ fecal score.

**Figure 2 microorganisms-12-00023-f002:**
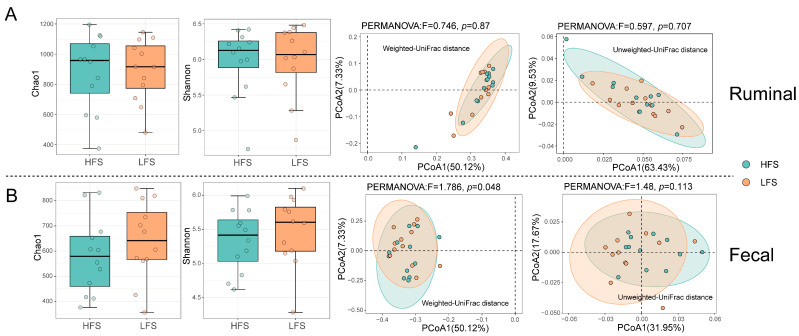
Ruminal (**A**) and feces (**B**) bacterial community diversity in dairy cows with different fecal scores. Wilcoxon Rank Sum Test was used to compare the difference in α diversity between the two groups, and a *p*-value with a false discovery rate correction < 0.05 was considered significant. Permutational multivariate ANOVA (PERMANOVA) based on weighted and unweighted UniFract distance was performed with 1000 permutations to test the difference in fecal bacteria β diversity.

**Figure 3 microorganisms-12-00023-f003:**
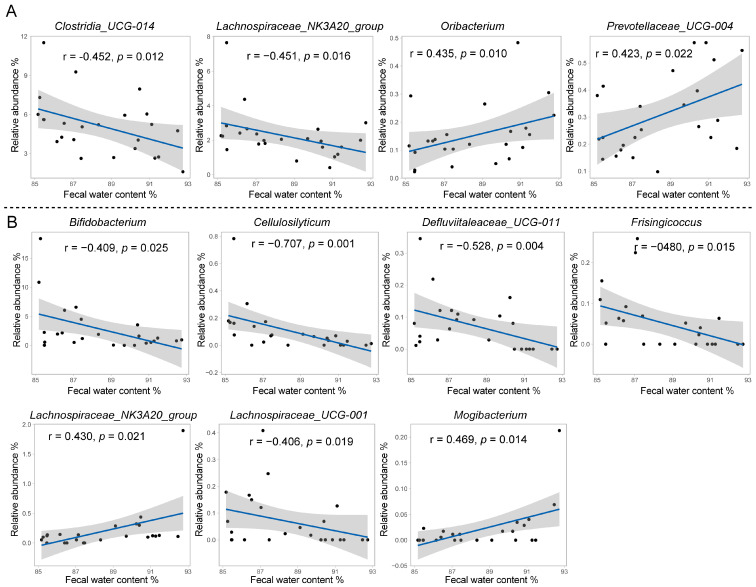
The Spearman rank correlation between the ruminal (**A**)/fecal (**B**) bacteria and fecal water content. The bacterial genera that were presented in at least 50% of the samples and had a significant correlation with fecal water content are presented here. Scatter plot with linear fit, the blue line indicates the best fit, and the gray area indicates 95% confidence interval. Each black dot represents a sample.

**Figure 4 microorganisms-12-00023-f004:**
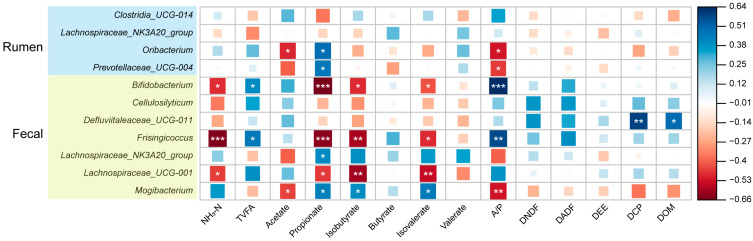
The Spearman rank correlation between the ruminal/fecal bacteria and ruminal/fecal fermentation profiles and apparent digestibility. The color denotes the correlation coefficient determined by the Spearman rank correlation. * *p* < 0.05, ** *p* < 0.01, *** *p* < 0.001. The microbial taxa in the light blue panel are ruminal bacteria that significantly correlated with fecal water content and were present in at least 50% of samples; the microbial taxa in the light yellow panel are fecal bacteria that significantly correlated with fecal water content and were present in at least 50% of samples. NH_3_-N: ammonia nitrogen; TVFA: total volatile fatty acid; A/P: acetate-to-propionate ratio; DNDF: digestibility of neutral detergent fiber; DADF: digestibility of acid detergent fiber; DEE, digestibility of ether extract; DCP: digestibility of crude protein; and DOM: digestibility of organic matter.

**Figure 5 microorganisms-12-00023-f005:**
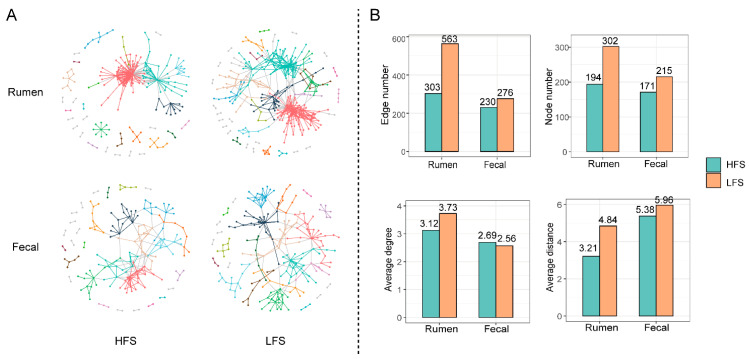
The visualization of constructed rumen and fecal bacterial networks (**A**) and network topology (**B**) in cows with different fecal scores. Large modules with ≥5 nodes are shown in different colors, and smaller modules are shown in grey.

**Figure 6 microorganisms-12-00023-f006:**
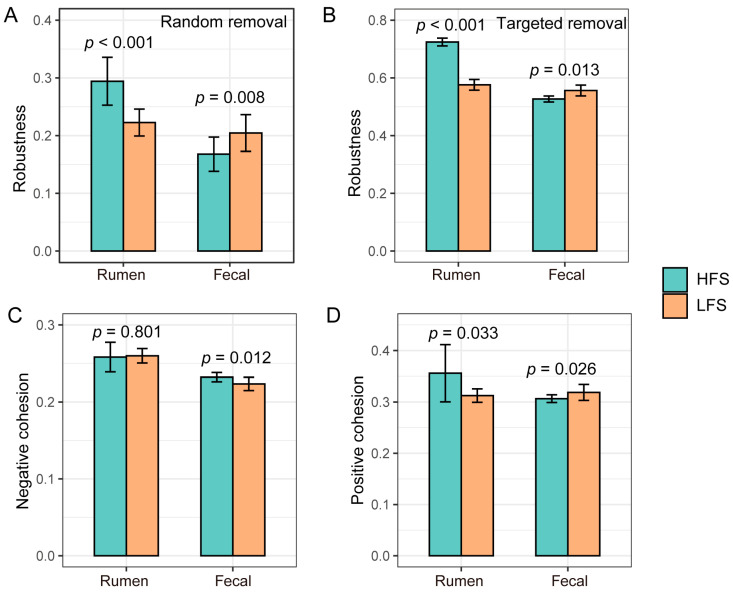
(**A**) Robustness was measured as the proportion of taxa that remained with 50% of the taxa randomly removed from each of the bacterial community networks. (**B**) Robustness was measured as the proportion of taxa that remained with three module hubs removed from each of the bacterial community networks. Negative cohesion (**C**) and positive cohesion (**D**) of ruminal and fecal bacterial communities in cows with different fecal scores.

**Table 1 microorganisms-12-00023-t001:** The rumen fluid and fecal fermentation profile of dairy cows with different fecal scores.

Items ^1^	Groups		SEM	*p*-Value
	LFS (*n* = 12)	HFS (*n* = 12)		
pH	6.54	6.59	0.05	0.668
NH_3_-N mg/dL	43.56	44.16	5.40	0.913
TVFA mM	83.37	77.90	4.77	0.266
VFA (mol/100 mol)				
Acetate	58.82	61.83	1.43	0.048
Propionate	26.13	22.07	1.38	0.008
Isobutyrate	0.45	0.53	0.08	0.359
Butyrate	12.16	13.06	0.61	0.155
Isovalerate	1.04	1.09	0.14	0.732
Valerate	1.41	1.43	0.14	0.868
A/P	2.32	2.85	0.20	0.016
Fecal fermentation profile				
NH_3_-N mg/g of feces	4.04	2.61	0.31	0.020
TVFA µmol/g of feces	63.14	81.06	5.23	0.002
VFA (mol/100 mol)				
Acetate	75.19	77.02	1.25	0.158
Propionate	13.33	12.11	0.64	0.069
Isobutyrate	1.33	0.82	0.15	0.003
Butyrate	7.76	8.40	0.98	0.525
Isovalerate	0.81	0.42	0.17	0.034
Valerate	1.57	1.24	0.16	0.054
A/P	5.71	6.49	0.37	0.046

^1^ LFS: low fecal score; HFS: high fecal score; NH_3_-N: ammonia nitrogen; TVFA: total volatile fatty acid; A/P: acetate/propionate; SEM: standard error of the mean.

**Table 2 microorganisms-12-00023-t002:** The fecal and rumen fluid, blood LPS, and blood binding protein content of different fecal score dairy cows.

Items ^1^	Groups		SEM	*p*-Value
	LFS (*n* = 12)	HFS (*n* = 12)		
Fecal LPS (EU/g of wet feces)	13,746.20	16,185.49	1049.20	0.260
Rumen fluid LPS (EU/mL)	14,255.22	12,208.48	1645.38	0.550
Blood LPS (EU/mL)	1.63	1.41	0.16	0.519
Blood LBP (mg/L)	2.01	2.51	0.16	0.117

^1^ LFS: low fecal score; HFS: high fecal score; LPS: lipopolysaccharides; LBP: LPS binding protein; SEM: standard error of the mean.

## Data Availability

The sequencing data are available in the NCBI Sequence Read Archive under project No. PRJNA973208.

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
