# Peer review of "Competitive Analysis of Rumen and Hindgut Microbiota Composition and Fermentation Function in Diarrheic and Non-Diarrheic Postpartum Dairy Cows"

_microorganisms, 2023, doi:10.3390/microorganisms12010023_

Round 1

Reviewer 1 Report

Comments and Suggestions for Authors

The manuscript is well structured and all used methods are adequately described. results are well presented. However, I would recommend some minor changes as following:  

Material and methods: 

It is not clear when cows were included in study. Namely, first samples were taken 9 days postpartum. However, I assume that cows were chosen and included in study some time before first sampling. 

Results: 

Line 223-you can choose to put exact p values or define all p values in the ranges you defined in 2.4. Statistics section

Line 247- use dots instead of comma in numbers present in results

Author Response

Dear Reviewer:

Thank you very much for your great suggestions and comments on my manuscript. I had revised the manuscript based on your high-level suggestions and replied to them point to point. Please check them in details as follows:

Point1: Material and methods: It is not clear when cows were included in study. Namely, first samples were taken 9 days postpartum. However, I assume that cows were chosen and included in study some time before first sampling.

Reply: Thank you very much for your suggestions. I agree with you that more detailed information should be provided for the animal used in this study. I added this information in the M&M section as follows:

Twenty-four multiparous Holstein dairy cows were enrolled in this study, with similar body weights (637 ± 28, mean ± standard error), parity (3.25 ± 0.14), body condition score (2.78 ± 0.19), and differences in the fecal score. The experimental period started after calving and ending at days in milk 19 days. And then the cows were divided into two groups, which were the low fecal score (LFS, n = 12), and high fecal score (HFS, n = 12) in the middle of the experimental period (d9). Fecal water content was also measured with a forced-air oven (DGG-9240B; Shanghai-ShenXin Inc, Shanghai, China) at 60°C for 48 h to further validate the fecal score accuracy.

Point2: Line 223-you can choose to put exact p values or define all p values in the ranges you defined in 2.4. Statistics section

Reply: Thank you very much for your detailed suggestions. I have added the      definition for the significant difference in the statistics section, which also revised in the manuscripts as follows:

The P-value ≤ 0.05 was set as significant and a tendency was considered as 0.05 < P-value ≤ 0.10 for all statistic analysis.

Point3: Line 247- use dots instead of comma in numbers present in results

Reply: Thank you very much for your suggestions. I had reformated the numbers and deleted the commas. The reformated sentences are as follows (also revised in the manuscript):

A total of 3079456 raw reads were obtained with an average of 65270 ± 8077 (mean ± standard deviation) reads per sample. After quality control, 756358 high-quality reads remained with an average of 15623 ± 2217 reads per sample. Good’s coverage was > 99% for each sample, indicating the sequence dept was sufficient to represent the bacteria community composition.

I deeply appreciate your suggestions and if there are any comments, please feel free to let me know, thank you in advance.

Best wishes

Yangyi Hao

Reviewer 2 Report

Comments and Suggestions for Authors

This article presents valuable information and intensive work to explore the reason behind the appearance of diarrhea in dairy cows. However, these points should be considered before making final decision.

Shortcomings: Sample collection: only one sample on day 9 was collected for milk, blood, and fecal analyses. This may be not enough to reflect animal status. Also, it is not clear when and how long animals  with diarrhea started to suffer the symptoms. So, it is important to show the time of symptoms appearance and the length of the symptoms, particularly relative to the time of samples collection and analyses.

Line 23: lower negative cohesion (competitive behaviors) and higher positive cohesion (co- operative behaviors)..Please, rephrase this sentence to be clearer,  how the same had both positive and negative cohesion???

Line 83: times and had free access to feed and water throughout the day…remove feed and (only water is offered all the day not the feed)

Lines 93-95: Twenty-four multip- arous Holstein dairy cows were enrolled in this study, with similar body weights, parity, body condition score, days in milk, and differences in the fecal score.. for each mentioned parameter show the mean and SE. Moreover, it is important to describe the nutritional regime applied on animals and if the diet was changed from one to another, particularly of animals are in the transition period. So, it is important to clearly mention the milking stage of animals and their nutritional regime to draw an overview on the possible reason of displaying of diarrhea in some animals.

Line 112: at -20 °C for apparent total tract….correct - to

Line 128: somatic cell count (SCC), BHBA, and acetone --I am not sure that these variables are analyzed using   an automated near-infrared milk analyzer .. Please, check and correct.

What is e ATTD, Do you mean apparent total tract digestibility (ATTP) ??? Use one abbreviation if you mean the same.

Line 131: About 25% of the mixed well fecal sample in each cow and TMR sample were dried .. Describe how these samples were collected and for how long for digestibility trial.

Comments on the Quality of English Language

English needs minor revision

Author Response

Dear Reviewer:

Thank you very much for your great suggestions and comments on my manuscript. I had revised the manuscript based on your high-level suggestions and replied to them point to point. Please check them in details as follows:

Point1: Line 23: lower negative cohesion (competitive behaviors) and higher positive cohesion (co- operative behaviors)..Please, rephrase this sentence to be clearer,  how the same had both positive and negative cohesion???

Reply: Thank you very much for your detailed information. I agree with you that our description is little bit confusing and I reworded the sentence as follows:

Fecal bacterial community in LFS cows had a higher robustness (P < 0.05) compared to HFS cows, which also had a lower negative cohesion (less competitive behaviors) and higher positive cohesion (more cooperative behaviors) (P < 0.05) compared to HFS cows.

In addition, we would like to explain little bit more about the cohesion. Based on the method reference that the negative cohesion and positive cohesion actually represented two different interactions mode in microbial community. To be specifically, negative cohesion, derived from negative pairwise correlations, could reflect the degree of competitive behaviors among the community members. Positive cohesion, derived from positive pairwise correlations, could reflect the degree of cooperative behaviors among the community members. To get to know the degree of competitive interactions or cooperative behaviors of ruminal and fecal bacteria communities in different fecal score cows, the negative and positive cohesions in each sample were calculated based on the abundance-weighted, null model-corrected positive and negative correlations, respectively [27]. I would like to provide the reference paper that publused in ISME J. We also calculated the two different cohesions in each sample based on the abundance-weighted, null model-corrected positive and negative correlations, respectively [27]. Therefore we descripted the positive and negative cohesion respectively.

Reference paper:

Herren CM, McMahon KD: Cohesion: a method for quantifying the connectivity of microbial communities. ISME J 2017, 11, 2426-2438. Https://doi.org/10.1038/ismej.2017.91.

Point2: Line 83: times and had free access to feed and water throughout the day…remove feed and (only water is offered all the day not the feed)

Reply: Thank you very much for your suggestions. Yes, the feed could be limited during the late of the day. I have deleted the fed and the revised sentence as follows:

All cows were fed (07:30, 14:30, and 20:30) three times and had free access to water throughout the day.

Point3: Lines 93-95: Twenty-four multip- arous Holstein dairy cows were enrolled in this study, with similar body weights, parity, body condition score, days in milk, and differences in the fecal score.. for each mentioned parameter show the mean and SE. Moreover, it is important to describe the nutritional regime applied on animals and if the diet was changed from one to another, particularly of animals are in the transition period. So, it is important to clearly mention the milking stage of animals and their nutritional regime to draw an overview on the possible reason of displaying of diarrhea in some animals.

Reply: Thank you very much for your suggestions. It is really useful for us to improve the quality of the M&M in this manuscripts. We do have the data about these animal information. I had added the mean and SE in the revised version and the revised sentence was as follows:

Twenty-four multiparous Holstein dairy cows were enrolled in this study, with similar body weights (637 ± 28, mean ± standard error), parity (3.25 ± 0.14), body condition score (2.78 ± 0.19), and differences in the fecal score (FS). The experimental period started after calving and ending at days in milk 19 days.

Also we provided the feed chemical composition and nutritional level in the supplement materials as Table S1. The experiment period is from after calving d1 to d19. So all the the experimental period was fed high-grain based diet as shown in Table S1. When the animals were involved in the study and the study period also clarified.

Point4: Line 112: at -20 °C for apparent total tract….correct - to -

Reply: Thank you very much for the suggestion. We had replaced the preposition “for” into “to”. We also double-checked the grammar related issues in other sections. Thank you very much.

Point5: Line 128: somatic cell count (SCC), BHBA, and acetone --I am not sure that these variables are analyzed using   an automated near-infrared milk analyzer .. Please, check and correct.

Reply: Thank you very much for your comments. I had double-checked with the lab. Yes, we used the FOSS (CombiFoss FT+; Foss Electric, Hillerød, Denmark) equipment and improted from Denmark. It can detected the milk SCC, BHBA and acetone indices. The introduction of the equipment could be see with the link: https://www.fossanalytics.com/en/.

Point6: What is e ATTD, Do you mean apparent total tract digestibility (ATTP) ??? Use one abbreviation if you mean the same.

Reply: Thank you very much for your suggestions. I had double checked and give the full name of the ATTD as follows:

About 300 g of feces was collected each time and stored at -20 °C to apparent total tract digestibility (ATTD) detection.

Point7: Line 131: About 25% of the mixed well fecal sample in each cow and TMR sample were dried .. Describe how these samples were collected and for how long for digestibility trial.

Reply: Thank you very much for your suggestions. We have added the fecal samples collection in the sample collection 2.2. We collected the fecal samples as follows described:

The fecal samples were collected from the rectum at 07:30, 14:30, and 20:30 on three consecutive days from d9 to d12 after calving. About 300 g of feces was collected each time and stored at -20 °C to apparent total tract digestibility (ATTD) detection.

In addition, for this compative study, all the experimental period is 19 days form after calving d1 to d19. Based on our previous observations, most of cows had the nutritional dairrhea symptoms from after d9. Therefore we collected the fecal samples from d9 after calving.

Point8: Shortcomings: Sample collection: only one sample on day 9 was collected for milk, blood, and fecal analyses. This may be not enough to reflect animal status. Also, it is not clear when and how long animals  with diarrhea started to suffer the symptoms. So, it is important to show the time of symptoms appearance and the length of the symptoms, particularly relative to the time of samples collection and analyses.

Reply: Thank you very much for your suggestions. We agree with you that it is really our shortcoming for this study. After calving, the introduction of high grain diet can cause a maladaptive issue of the gastrointestinal tract in postpartum dairy cows and as a result, cows can develop nutritional diarrhea. Although many cows can recover eventually, some of them need to be treated or culled due to the severity of nutritional diarrhea. Also the period of cows with diarrhea symptoms also had a high individual variations. Based on our competitive analysis, the gut microbiota and host performance, such as blood parameters and milking performance do different between the two groups cows. However, with these diarrhea cows, when and how long the diarrhea companied with the cows is really a interteting questions and it really deserved to collect data further.

I deeply appreciate your suggestions and if there are any comments, please feel free to let me know, thank you in advance.

Best wishes

Yangyi Hao

Reviewer 3 Report

Comments and Suggestions for Authors

The manuscript shows interesting results regarding gastrointestinal microbiota and fermentation limits, which could contribute to ruminant health and welfare

LN25: P

LN 65-75: Suggested to rewrite clearer.

LN 89: fecal score (FS)

LN 95:  FS

Revise the abbreviation in the manuscript.

Suggested to avoid the terms “We, I, he...” in the entire manuscript.

Enhance the quality of figures 1, 2 and 3.

Overall, the experimental protocol is well described and the methods are adequate including the statistical analysis.

Suggested proofreading to avoid a few syntax errors.

In conclusion – use the full form for abbreviated words. Ex. LFS, HFS, LPS

This reviewer does not have any negative comments on the manuscript.

Comments on the Quality of English Language

Suggested proofreading to avoid a few syntax errors. 

Author Response

Dear Reviewer:

Thank you very much for your great suggestions and comments on my manuscript. I had revised the manuscript based on your high-level suggestions and replied to them point to point. Please check them in details as follows:

Point1: LN25: P

Reply: thanks you very much for your comments. We had revised the p into P in the revised version.

Point2:LN 65-75: Suggested to rewrite clearer.

Reply: Thank you very much for your suggestions. We had rewrote this part and as follows.

The hypothesis is that the sudden introduction of a high-energy, high-protein diet to postpartum dairy cows could lead to dysbiosis in the microbiota of the rumen and hindgut, impacting their metabolites and resulting in nutritional diarrhea. Specifically, the abrupt increase in a high-grain diet might elevate lipopolysaccharides (LPS) levels and alter microbial protein degradation and fermentation in the gut, which further lead to nutritional diarrhea in postpartum dairy cows. In this study, the cows exhibiting high or low fecal scores (indicative of nutritional diarrhea) were seletced and their milking performance, ruminal and fecal fermentation profiles, ruminal, fecal, and blood LPS levels, apparent total tract digestibility of feed, as well as the composition of rumen and fecal bacteria were compared. The aim was to elucidate the factors contributing to nutritional diarrhea in postpartum dairy cows on commercial dairy farms.

Point3:LN 89: fecal score (FS)

LN 95:  FS

Revise the abbreviation in the manuscript.

Suggested to avoid the terms “We, I, he...” in the entire manuscript.

Reply: Thank you very much for your suggestions. I had double-checked all the abbrevised and also deleted the abbreviation of FS for it is too short to give abbreviations.

Point4:Enhance the quality of figures 1, 2 and 3.

Reply: Thank you very much for your suggestions. We do have the PDF version of these figures. First we replaced the figures in the manuscript with high-quailty figures. And we also will send to the editior the PDF version of figures.

Point5: In conclusion – use the full form for abbreviated words. Ex. LFS, HFS, LPSI

Reply: Thank you very much for your suggestions and we have gave the full name of these abbreviations in the conclusion parts. The revised version is as follows:

This study revealed differences in gastrointestinal fermentation profiles, as well as gastrointestinal bacteria composition between diarrhetic and non-diarrhea postpartum dairy cows. No difference in ruminal pH, total volatile fatty acids, and lipopolysaccharide indicates that nutritional diarrhea in the postpartum dairy cows was not caused by rumen acidosis and/or lipopolysaccharide. Lower fecal total volatile fatty acids and higher ammonia nitrogen concentration in low fecal score cows suggest that nutritional diarrhea in the postpartum dairy cows in this study could be caused by the low carbohydrate and high protein fermentation in the lower gut as well as the gut dysbiosis. Key fecal bacterial genera such as Bifidobacterium, Frisingicoccus, and Lachnospiraceae_UCG-001 were identified as significantly correlated with fecal water content and fermentation profiles which could maintain the homeostasis in the hindgut. Besides, the low fecal scores cows fecal bacterial community had a lower adaptability to diet shift compared to high fecal score cows, which could affect the microbial composition and fermentation in the hindgut. Metagenomes and metabolomes are required in the future to assess the microbial function and their metabolites in the gut to uncover the mechanism of postpartum dairy cows’ nutritional diarrhea. Regardless, the findings from this study have provided knowledge on the importance of shaping hindgut bacteria composition and function in the lower gut of postpartum dairy cows, which could be one of the directions to improve gut health and reduce nutritional diarrhea in postpartum dairy cows.

We deeply appreciate your suggestions and if there are any comments, please feel free to let me know, thank you in advance.

Best wishes

Yangyi Hao

Round 2

Reviewer 2 Report

Comments and Suggestions for Authors

Twenty-four multiparous Holstein dairy cows were enrolled in this study, with similar body weights (637 ± 28, mean ± standard error), parity (3.25 ± 0.14), body condition score (2.78 ± 0.19), and differences in the fecal score.

= Mention the unit of each variable…kg, year,……

Comments on the Quality of English Language

English needs minor revision

Author Response

Dear Reviewer:

Thank you so much for your suggestions again. We really appreciate it. I have revised the animal information in M&M section as follows (the green part in the manuscript):

Twenty-four multiparous Holstein dairy cows were enrolled in this study, with similar body weights (637 ± 28 kg, mean ± standard error), ages (5.34 ± 0.27, years), body condition score (2.78 ± 0.19), and differences in the fecal score.

For the body weight, the unit is kg, and the age is years. However, for the parity and body condition scores, there is actually no unit for they were numeric.

I deeply appreciate all your detailed and high-level comments. Thank you so much.

Best wishes

Yangyi Hao
